# Spatial Interaction Analysis of Infectious Disease Import and Export between Regions

**DOI:** 10.3390/ijerph21050643

**Published:** 2024-05-18

**Authors:** Mingdong Lyu, Kuofu Liu, Randolph W. Hall

**Affiliations:** 1National Renewable Energy Laboratory, Mobility, Behavior, and Advanced Powertrains Department, Denver, CO 80401, USA; 2Epstein Department of Industrial and Systems Engineering, University of Southern California, Los Angeles, CA 90089, USA; kuofuliu@usc.edu (K.L.); rwhall@usc.edu (R.W.H.)

**Keywords:** interregional travel, disease transport, healthcare policy, compartmental model, COVID-19

## Abstract

Human travel plays a crucial role in the spread of infectious disease between regions. Travel of infected individuals from one region to another can transport a virus to places that were previously unaffected or may accelerate the spread of disease in places where the disease is not yet well established. We develop and apply models and metrics to analyze the role of inter-regional travel relative to the spread of disease, drawing from data on COVID-19 in the United States. To better understand how transportation affects disease transmission, we established a multi-regional time-varying compartmental disease model with spatial interaction. The compartmental model was integrated with statistical estimates of travel between regions. From the integrated model, we derived a transmission import index to assess the risk of COVID-19 transmission between states. Based on the index, we determined states with high risk for disease spreading to other states at the scale of months, and we analyzed how the index changed over time during 2020. Our model provides a tool for policymakers to evaluate the influence of travel between regions on disease transmission in support of strategies for epidemic control.

## 1. Introduction

Travel plays a crucial role in the spread of infectious disease between regions. COVID-19, for example, can be spread through respiratory droplets that are released when an infected person talks, coughs, or sneezes. These droplets can then be inhaled by other individuals in proximity to the infected person. Transportation modes such as buses, trains, and airplanes are high-risk areas for the transmission of the virus, as they often involve large numbers of people in enclosed spaces for extended periods of time [1]. In addition, long-distance travel can significantly impact the spread of diseases to new areas. When people travel long distances, they can bring infectious agents with them. These agents can then spread to new populations and new areas, potentially causing outbreaks of disease.

Whereas COVID-19 originated in a specific location in China in late 2019, human travel provided a vector through which it spread from region to region. As illustrated in Figure 1, regions may export the disease to adjacent regions through travel of infected individuals from adjacent regions or from long-distance travel of infected individuals from far away. Once the disease has been imported to a region, it may subsequently be transmitted among individuals within the region via “community spread” and may also be exported elsewhere through travel of infected individuals, either to places where the disease is not yet present or back to places where the disease is already present (second phase of Figure 1). In the third phase of Figure 1, community spread can become the dominant form of infection and all regions, now experiencing infections, will both import and export disease. Over time, the relative risk of contracting the disease via travel of infected individuals from elsewhere versus community spread will change within regions. On the other hand, if a region has, through public health measures, greatly reduced the prevalence of disease, the risk from importation may become the dominant—or perhaps the only—risk for developing new infections. If the region aims to maintain low risk of disease, it may need to prohibit travel into the region, or impose quarantines on incoming travelers, as occurred in China. 

In sum, travel restrictions may be effective at disease control, but their effectiveness depends on the prevalence of disease within the region, the prevalence of disease elsewhere, and the volume of travel between regions. Therefore, studying the impact of travel on the transmission of diseases is crucial for understanding how infectious diseases can spread across different regions and populations. Studying the impact of travel on disease transmission can also inform broader discussions about global health and the interconnectedness of populations around the world. As travel has become common and widespread, it is increasingly important to understand how diseases can be transmitted between regions and how to prevent the spread of infectious agents.

This paper aims to investigate the transmission of diseases through spatial interaction, with a specific focus on state-level travel in the United States. We develop and apply a transmission import index related to transportation to assess the impact of long-distance travel on disease transmission. By gaining a better understanding of the effects of travel on disease transmission, we can analyze historical disease outbreaks and develop effective strategies for preventing and controlling future outbreaks and pandemics. 

## 2. Literature Review

Traveler behavior underlies the spread of infectious diseases between regions. Various modes of transportation, such as air [2,3,4], rail [5], and water [2], can facilitate the spread of infectious diseases. In epidemiology, understanding the impact of human mobility on transmission dynamics is needed for disease modeling and investigation. Spatial interaction models of human mobility have been employed to study epidemics using two approaches: modeling spatial interactions between distinct population groups and integrating spatial interaction into epidemic models.

### 2.1. Spatial Interaction Models in Epidemics

The gravity model [6] and radiation model [7] are widely used spatial interaction models for analyzing travel between regions as well as the spread of infectious diseases. The gravity model is widely used in transportation modeling due to its simplicity and ability to predict transportation flows between regions. The gravity model has been applied in a wide range of transportation contexts, including freight transportation, passenger transportation, and tourism. The model has also been applied to various transportation modes, including air, sea, and land transportation. Additionally, the gravity model can be adapted to include other variables that may influence transportation flows, such as population density, income, or trade barriers.

The gravity model postulates that travel between regions is positively correlated with the product of region sizes (such as populations or gross domestic products, GDP) and inversely proportional to the square [6] or non-quadratic of the inter-region distance. Because each person traveling from one region to another is potentially infectious, the rate at which new infections occur due to travel is proportionate to the predicted number of trips. The gravity model has been used to examine the transmission of influenza between regions in Bangladesh [8], population centers in England and Wales [9], Mexico and major global cities [10], and states in America [6,11]. Additionally, the gravity model has been applied to other infectious diseases transmission. Xia et al. embedded a metapopulation-based gravity coupling model in a time series susceptible–infected–recovered (TSIR) model to simulate measles dynamics in England and Wales [12]. Barrios et al. used the gravity model to study the spatial spread of vector-borne diseases, including nephropathia epidemica and lyme borreliosis between the physical habitat of pathogens and urban areas in Belgium [13]. A limitation of the gravity model is that it may not accurately depict regions characterized by significant heterogeneity and uncertainty [14]. It also requires a tuning process to estimate the model parameter based on real observations.

As an alternative, Simini proposed the radiation model to describe mobility patterns only using the population data between regions [7]. Kraemer et al. introduced a versatile transmission model to assess the effectiveness of generalized human movement models, including the radiation model, in estimating cases of Ebola virus disease (EVD) and mapping the spatial progression of the outbreak [15]. Tizzoni et al. utilized the radiation model to investigate the spread of influenza-like-illness (ILI) epidemics between a set of European countries [16]. Kraemer et al. used a logistic formula incorporating parameters from the gravity model and the radiation model to simulate the yellow fever virus outbreak in Angola and the Democratic Republic of the Congo [17].

Compared to the gravity model, Kang et al. and Masucci et al. found that the radiation model was more accurate for long-distance travel [18,19]. However, the radiation model only takes into account large-scale parameters. Thus, the radiation model has relatively poor accuracy on short-distance travel [19]. The gravity model and radiation model both involve detailed data regarding inter-regional travel for parameter estimation, which is a challenge due to the complexities associated with collecting mobility data with precision. Furthermore, privacy regulations regarding highly detailed mobile data pose an obstacle [16], impeding the establishment and application of spatial interaction models utilizing different datasets [8].

### 2.2. Epidemic Models with Spatial Interaction

Besides direct modeling of spatial interaction, another approach for modeling inter-regional disease transmission is epidemic models incorporating spatial interaction terms. In the 1980s, geographers first proposed a spatial framework for epidemiological models that explicitly considers the spatial dispersion of infectious diseases. A simple form of these spatial models is the three waves model projecting the infectious, susceptible, and recovery populations in a two-dimensional grid [20,21]. The population-based wave model has been applied to pandemic waves in a large space, such as the 1918–1920 Spanish flu [22]. 

Instead of assuming the whole population as an identical model unit with three waves, spatially structured models can divide the population into a substantial number of subpopulations that are homogeneous within each group and heterogeneous from each other [21,23,24]. Spatially structured models can be derived with epidemiological compartment models [25], which divides the population in one region into three categories: susceptible, infectious, and recovered phases, combining with the spatial interactions between several investigated regions. Such compartment models provide insights for understanding infectious disease dynamics. Vrabac et al. proposed a transportation network embedded Susceptible–Exposed–Infectious–Recovered (SEIR) model and applied the model to simulate COVID-19 transmission between 110 counties in the United States [26]. Kuzdeuov et al. developed a network-based simulator to account for the effect of transportation of COVID-19 spreading among 17 administrative regions in Kazakhstan [27]. Levin et al. investigated the patterns of COVID-19 transmission, including short-term travel between counties in Minnesota with a modified SEIR model [28]. Hatami et al. studied the spatio-temporal dynamics of COVID-19 in 10 counties in the Charlotte–Concord–Gastonia Metropolitan Statistical Area [29]. The spatially structured model is also effective when implemented in mobile and high-density populations, such as military or refugee camps [30,31]. As a further exploration of the spatial structured model, individual-based spatial models divide the subpopulation into individual compartments [22]. Eames and Keeling developed pair-wise network equations utilizing the essential characteristics of the mixing network to estimate the effectiveness of various control strategies towards sexually transmitted diseases. However, individual-based models require a great amount of detailed information about individuals, which is generally not available, especially at the early stage of an epidemic when inter-regional spread is particularly important to predict.

### 2.3. Time-Varying Models

Simulating infectious disease transmission with the SEIR model deepens the understanding of disease dynamics. Compartment models have been useful in modeling cases of H1N1 [32], COVID-19 [29,33], and measles [34] in the United States. Prior studies generally applied a standard SEIR model, which adopts a constant basic reproduction number. However, the transmissibility of each virus evolves with time, leading to the change in reproduction number [35]. Constant parameters in the standard SEIR model fail to capture the evolving nature of infectious viruses, such as influenza. Moreover, various government interventions like social distancing, masking, and vaccine administration can also affect the transmission of diseases. Compared to the standard SEIR model, a time-varying SEIR model is more accurate in depicting infectious disease transmission. Wang et al. proposed a constrained time-varying SEIR model to explore optimal vaccine allocation strategies and achieve a superior result compared to the standard model [36]. Feng et al. developed an algorithm combining deep learning and the SEIR model with time-varying parameters to predict COVID-19 cases in the United States [37]. Despite the complexity of time-varying parameters, incorporating spatial interaction into epidemiological models is essential for capturing disease dynamics, particularly in the early phases of a pandemic [38]. Nevertheless, prior studies have not used a time-varying SEIR model to investigate inter-regional COVID-19 transmission at the state level in the United States. Such a model could be important for vaccine allocation [39] or state-level travel restrictions, given the importance of states as governance units in the United States. Additionally, quantitative evaluation of the impact of transportation on infectious disease transmission poses challenges for state governments due to the absence of appropriate indices. 

In the following sections, we both define our model for disease transmission between regions and define an index to assess the importance of inter-regional travel for the spread of disease. We then apply the model to the United States, using the 50 states as regions, covering the time period from March 2020 to September 2020. Last, we assess the extent to which interstate interactions affected disease spread by date and assess which states played the biggest role in exporting disease to other states as a function of time.

## 3. Sources and Methodology

We describe the data source and the proposed compartment model in this section. Based on the traditional SEIRD compartment model and a time-varying modification in the parameters, we introduce the transportation impact into the mathematical model for our analysis. We utilize a combination of two data sources to estimate the regional travel volumes on a daily scale and then calibrate the model with COVID-19 case and mortality data. We interpret the fitting results with our index for assessing the relative importance of disease import from other states versus community spread from within the states. We did not analyze the import and export of disease among nations.

### 3.1. Data Sources

This section introduces our primary data sources for estimating human travel between regions: the “Trips by Distance” data from the Bureau of Transportation Statistics and the COVID-19 Impact Analysis Platform by the Maryland Transportation Institute (MTI) and Center for Advanced Transportation Technology Laboratory (CATT Lab). By combining these sources, we collect and calculate the daily out-of-state trips for each state for model input.

#### 3.1.1. Trips by Distance Data

The Bureau of Transportation Statistics (BTS) collected and curated data on the number of trips taken in the United States by distance, mode of transportation, and purpose of trip [40]. The data are available for the years 2019 to 2022, and the daily travel estimates are based on a merged mobile device data panel that addresses issues with geographic and temporal variation.

Trips are defined as movements that include a stay of longer than 10 min at an anonymized location away from home, and the data capture travel by all modes of transportation. A movement is considered to consist of multiple trips when it includes multiple stops, each lasting more than 10 min. The data are presented by the BTS at the national, state, and county levels, and a weighting procedure is used to ensure the sample of mobile devices is representative of the entire population in each area. To protect confidentiality and support data quality, the source does not include data for a county if there are fewer than 50 devices in the sample on any given day. The dataset is combined with the following dataset provided by the COVID-19 Impact Analysis Platform to generate the spatial interaction flow.

#### 3.1.2. COVID-19 Impact Analysis Platform

The COVID-19 Impact Analysis Platform [41] was developed by the Maryland Transportation Institute (MTI) and Center for Advanced Transportation Technology Laboratory (CATT Lab). The platform provides a range of data and analytical tools, including interactive maps, visualizations, and dashboards, to help users better understand the spread of the virus and its impact on various social and economic indicators.

The platform integrates multiple data sources, including public health data, mobility data, and socioeconomic data, to provide a more comprehensive picture of the pandemic’s impact. Specifically, the mobility data track daily visits to different types of locations, such as retail and recreation areas, transit stations, workplaces, and grocery stores, and compares them to those of pre-pandemic levels. The mobility data are derived from anonymized and aggregated data from mobile devices, such as smartphones and tablets, that have opted into location tracking services. The data are aggregated at the county level in the United States. For the analysis of spatial interactions, the platform specifically provides the state/county level percentage of out-of-state/out-of-county trips per day from 1st January 2020 to 30 April 2021.

We combined the mobility data provided by the COVID-19 Impact Analysis Platform and the daily trips from the BTS to fit our model. However, it is important to note that the mobility datasets are based on a sample of mobile devices and may not be representative of the entire population and that the datasets are anonymized and aggregated to protect user privacy.

#### 3.1.3. Disease Data

Daily cases and deaths by state were utilized in our compartmental disease model, as described in Section 3.3. Data were obtained from the COVID-19 tracking project led by *The Atlantic* (derived from the Centers for Disease Control), for each 30-day period between 15 March 2020 and 15 October 2020.

### 3.2. Estimation of Travel between Regions

Our source data, discussed in Section 3.1, provide estimates of total daily out-of-state trips for each state, but do not estimate trips by destination state. In this section, we apply the gravity model to estimate the distribution of trips among states on a daily basis throughout the investigated period. 

We utilized the following form of the gravity model [9], where region size is defined by GDP [42]:(1)$$M_{ij}=M_{i}*\frac{\frac{G_{j}^{\alpha }}{D_{ij}^{\gamma }}}{\sum_{k}\frac{G_{k}^{\alpha }}{D_{ik}^{\gamma }}}$$
where $M_{ij}$ represents the flow of trips from region i to region j, $M_{i}$ is the total flow of trips out of region i, $G_{k}$ is the GDP of region k, $D_{ik}$ represents the distance between the region i and region k, and α and γ are exponents that determine the relative influence of the variables. $\frac{G_{j}^{\alpha }}{D_{ij}^{\gamma }}$ represents the attraction index of region j calculated by the gravity model. For distance, we calculated centroid-to-centroid distances for all pairs of regions, excluding pairing each region to itself. The ratio $\frac{\frac{G_{j}^{\alpha }}{D_{ij}^{\gamma }}}{\sum_{k}\frac{G_{k}^{\alpha }}{D_{ik}^{\gamma }}}$ shows the proportion of the total trips from region i to region j. Because real trip flow data from region i to region j were not available, we could not statistically estimate the GDP power-law parameter α and the distance power-law parameter γ. We have instead analyzed the scenarios of setting α at values of 0.5, 1, 1.5, and 2, respectively, while keeping γ fixed at 2, as well as assigning γ to be 0.5, 1, 1.5, and 2 with α fixed at 1.

In general, movement of people between regions makes it possible for infected people to transmit the disease to susceptible people residing at their destinations. Thus, it creates the potential for the disease to spread between regions. As predicted by the gravity model, states that are adjacent or otherwise geographically close to each other tend to have more spatial interaction compared to states that are far apart. Larger states also produce more travel, creating more potential to spread the disease to other states.

### 3.3. Disease Transmission Model

In this section, we introduce our compartmental model for disease transmission, which represents both community spread of disease from infectious people residing in a particular region and importation of disease from infectious people traveling from other regions. Our estimates of daily trips between regions were used as a model input. The model is defined as follows:(2)$$\begin{array}{l} \frac{\partial S_{i}\left(t\right)}{\partial t}=-\beta_{i}\left(t\right)\cdot I_{i}\left(t\right)\cdot \frac{S_{i}\left(t\right)}{N_{i}}+\sum_{j\neq i}\mu_{j}(t)\sum_{j\neq i}\frac{M_{ij}(t)S_{j}}{N_{j}-I_{j}}-\mu_{i}(t)\sum_{j\neq i}\frac{M_{ji}{(t)S}_{i}}{N_{i}-I_{i}}\\ \frac{\partial E_{i}\left(t\right)}{\partial t}=\beta_{i}\left(t\right)\cdot I_{i}\left(t\right)\cdot \frac{S_{i}\left(t\right)}{N_{i}}-\sigma \cdot E_{i}\left(t\right)+\sum_{j\neq i}\mu_{j}(t)\sum_{j\neq i}\frac{M_{ij}(t)E_{j}}{N_{j}-I_{j}}-\mu_{i}(t)\sum_{j\neq i}\frac{M_{ji}{(t)E}_{i}}{N_{i}-I_{i}}\\ \frac{\partial I_{i}\left(t\right)}{\partial t}=\sigma \cdot E_{i}\left(t\right)-\left(1-\alpha_{i}\left(t\right)\right)\cdot \gamma \cdot I_{i}\left(t\right)-\alpha_{i}\left(t\right)\cdot \rho \cdot I_{i}\left(t\right)\\ \frac{\partial R_{i}\left(t\right)}{\partial t}=\left(1-\alpha_{i}\left(t\right)\right)\cdot \gamma \cdot I_{i}\left(t\right)\\ \frac{\partial D_{i}\left(t\right)}{\partial t}=\alpha_{i}\left(t\right)\cdot \rho \cdot I_{i}\left(t\right) \end{array}$$
where $S_{i}\left(t\right),E_{i}\left(t\right),I_{i}\left(t\right),R_{i}\left(t\right),D_{i}\left(t\right)$ and $N_{i}$ are the susceptible, exposed, infected, recovered, dead, and total population in region i at time t. $M_{ij}(t)/M_{ji}(t)$ represents the flow of trips from region j/i to region i/j at given time t. The transformation rates in an epidemiological model are denoted as σ, γ, and ρ, where σ is the rate from exposed to infectious, equivalent to the reciprocal of the incubation period; γ is the rate from infectious to recovered, corresponding to the reciprocal of the recovery time; and *ρ* is the rate from infectious to dead. The values of γ, σ, and *ρ* are set to 1/6.5 [43], 1/3.0 [43], and 1/7.5, respectively [44]. $\beta_{i}\left(t\right)$ and $\alpha_{i}\left(t\right)$ are time-varying parameters following a Sigmoid function-based form [45] representing the reproduction number and fatality rate, respectively, at time t in region *i*. Spatial interaction between regions is represented by the daily number of people traveling from region j to region i and an adjustable factor $\mu_{i}(t)$ denoting rate at which an infectious traveler from region *i* transmits disease to individuals in other regions per day, at time t. 

We use the Standard Federal Regions [46] to aggregate the 50 states into 10 parts. The Standard Federal Regions are based on geographic, economic, and cultural factors. They were designed to promote efficient and effective delivery of federal programs and services by bringing together federal agencies, state and local governments, and private organizations to work collaboratively and address regional issues and concerns. The states within the same region have proven to be able to share resources, expertise, and best practices across state lines and jurisdictions. 

To reduce model complexity, we assume that states in the same region share similar traits of geographic, economic, and cultural factors. For instance, potential travelers in the same region may have similar destinations and purposes of traveling. Thus, a common μ value, which represents potential infection spread by an infectious traveler from region *i*, was assigned to states under the same division. We employ the Levenberg–Marquardt algorithm (LMA) within Python 3.10.0 to fit the model using both pandemic and transportation data.

### 3.4. Transmission Import Index

From Equation (2), the term $\sum_{j\neq i}\mu_{j}(t)\sum_{j\neq i}\frac{M_{ij}(t)E_{j}}{N_{j}-I_{j}}$ represents the rate at which people become newly exposed in region *i* at time t due to the import of disease from other regions. Meanwhile, the term $\beta_{i}\left(t\right)\cdot I_{i}\left(t\right)\cdot \frac{S_{i}\left(t\right)}{N_{i}}$ is the rate at which people become newly exposed in region i due to local transmission. The total rate at which people become newly exposed in region *i* at time t is the sum of these two rates.

To evaluate the relative risk of disease transmission due to import from other regions, we introduce the transmission import index: (3)$$Transmission\;Import\;Index=\frac{\sum_{j\neq i}\mu_{j}(t)\sum_{j\neq i}\frac{M_{ij}(t)E_{j}}{N_{j}-I_{j}}}{\beta_{i}\left(t\right)\cdot I_{i}\left(t\right)\cdot \frac{S_{i}\left(t\right)}{N_{i}}+\sum_{j\neq i}\mu_{j}(t)\sum_{j\neq i}\frac{M_{ij}(t)E_{j}}{N_{j}-I_{j}}}$$

The numerator is the rate of newly exposed people due to travel from other regions into region *i* at time t. The denominator is the total rate of newly exposed people in region *i* at time t. The transmission import index quantifies the impact of travel into region *i* relative to the total rate of new exposures. If the number of travelers from all other regions is equal to 0, the numerator will be 0, resulting in the index also being 0. This signifies that import has no impact on the spread of the disease in region *i* at time t. On the other hand, if there are currently no people exposed to disease in region *i*, then the import index either equals 1 (if there is travel of exposed people into the region) or is undefined (if there is no travel of exposed people into the region). 

The transmission import index offers insights into the degree to which a region’s disease transmission is impacted by travel from other regions. A higher transmission import index for region *i* indicates a greater influence of incoming individuals on infectious conditions in that region. Consequently, policymakers in such regions might opt to enforce stringent regulations limiting travelers from other areas or implement quarantine measures to mitigate the heightened risks of disease spread. Conversely, if the index is low, restrictions on incoming travelers may not yield significant benefits, as the greater risks stem from community spread within the region.

The transmission import index is an important metric for assessing the potential risk of disease transmission that one region experiences through multi-regional spatial interaction. By considering the effect of travel from other regions, this index serves as a valuable guide for healthcare policymakers. Furthermore, the change in the transmission import index over time could indicate the trend of disease transmission and help policymakers assess the impact of interventions. 

## 4. Analysis and Results

The results of our application of the compartment model to the United States are illustrated in this section. We analyzed domestic transmission of disease among states and not international transmission between nations. In Section 4.1, we provide fitting results, showing root-mean-squared errors. We provide the transmission import index results in Section 4.2 to quantify the impact of travel on disease transmission, by location and date.

### 4.1. Fitting Results

We fit the model with the dataset of 7-day moving average cases and deaths for the 50 states, provided by the COVID-19 tracking project lead by *The Atlantic* (derived from the Center for Disease Control), for each period of 30 days from 15 March 2020 to 15 October 2020. The fitting accuracy across all states with γ set to be two and α to be one is presented in Figure 2, measured by the relative root mean square error (RRMSE) defined as $RRMSE=\frac{{\left[\sum_{i=1}^{N}{\left(\hat{y_{i}}-y_{i}\right)}^{2}/N\right]}^{1/2}}{y_{N}}$.

The average RRMSE of the reported cases over 7 months ranges from 0.54% to 3.78% and of the reported deaths ranges over 7 months from 0.24% to 2.49%. The average and median RRMSEs for cases are 1.54% and 1.48%; for deaths, the average and median values are 1.20% and 1.14%. Figure 2 shows the average RRMSE for cases and deaths over 7 months across 50 states of the dynamic model with multi-regional spatial interaction. 

Figure 3 and Figure 4 display the fitted results for COVID-19 cases and deaths in example states (Georgia, New Jersey, Florida, and Maryland, USA), during the period from 15 October 2020 to 15 November 2020.

In summary, the dynamic modeling with multiregional spatial interaction demonstrates a high degree of accuracy in capturing the historical transmission dynamics of infectious diseases. This method effectively accounts for the complexities and interactions between various regions, leading to a more comprehensive understanding of the factors influencing disease spread and the effectiveness of control measures.

### 4.2. Transmission Import Index

Figure 5 shows the fluctuation of the transmission import index in five states from mid-March to mid-September 2020. Each time period ends on the 15th of a month. For example, we refer to the April time period as the period ending on April 15. From March to June, the average index among the states experienced a decline, potentially attributable to both a decline in domestic travel and the wider prevalence of the disease throughout the nation (as a consequence of prior importation). Throughout the investigation period, Illinois consistently exhibited indices higher than the national average, whereas Texas consistently maintained lower indices compared to the US average. We note that Illinois has traditionally been a hub for domestic travel in the US, offering greater potential for disease import. On the other hand, Texas is surrounded by low population states, reducing interstate travel. 

New York’s index exhibited two surges, from April to May and later from July to August. In contrast, the California index declined from March to April but increased from August to September, remaining stable in the interim. Florida initially saw a decrease in the index during the first month under investigation, stabilizing in the subsequent five months. The magnitude and variation of the transmission import index within a state may be connected to the state’s characteristics, including size and proximity to other states, and social events that can cause interstate travel. Given that the outbreak’s first wave included many infections in New York State, local transmission dominated import from other states in the first time period. As the disease spread to surrounding states, New York become more susceptible to disease import.

In contrast, California experienced a high import index in April, as local spread was initially minimal. By May, as interstate travel declined and local spread increased, the important index became very small. After the initial outbreak, the transmission import index for Illinois and New York remained consistently higher than the United States average. For California, Florida, and Texas, the transmission import index remained consistently lower than the United States average. This trend may be attributed in part to the vast and varied geographic landscapes of California, Florida, and Texas, which include both densely populated urban areas and sparsely populated rural regions around state boundaries, mitigating the effects of travel on the disease transmission. In contrast, New York City, the major population center in New York State, is adjacent to two other states (New Jersey and Connecticut) and less than 100 miles from Pennsylvania. Fluctuations in the transmission import index may be affected by events that attract visitors from adjacent states. Between July and August, the transmission import index exhibited an upward trend for New York and Illinois, indicating a growing impact of travelers on disease transmission. This trend may be connected to the quarantine policy implemented in New York City in early August [47] and the quarantine order for 15 states issued in Chicago in July [48], which affected local transmission.

Figure 6 summarizes the trend of average transmission import indices in the top ten population states and the bottom forty population states from March to September 2020. States with larger populations consistently exhibited higher index values than both the US average and the remaining lower-population states. More populous states are typically developed regions, naturally attracting a higher number of travelers passing through their boundaries. These individuals could potentially act as carriers of diseases, increasing the risk of disease transmission. Overall, the average transmission import index decreased from March 20th to July 15th and increased from July 15th to September 15th for the top ten, bottom forty states, and all fifty states in the US. The initial outbreak is reflected in the high index for April 2020. From March 20th to July 15th, the average transmission import index decreased for all groups. This decline may be linked to the initial impact of public health measures restricting travel after the pandemic’s onset and the increased occurrence of local transmission. From July to August, the index increased as travel resumed, spreading disease to areas that had previously seen a decline in rates of new cases. Subsequently, from August to September, the average transmission import index decreased again, including both the top ten highest-populated and the bottom forty least-populated states. The decline in the index may be linked to a resurgence of local transmission of the disease [49].

Figure 7 and Figure 8 display heat maps for the normalized transmission import index, which ranges from zero to one. In this context, zero signifies the state with the lowest import index, while one indicates the state with the highest import index. All other states are proportionately assigned to values between 0 and 1; Figure 7 depicts the normalized import index at the time of the initial outbreak for the April period. The northeast region, encompassing states such as Vermont and Massachusetts displayed notably high transmission import indices, signifying a heightened risk of travel from other states, such as New York. The heat map aligns with early introduction of COVID-19 in early 2020 when northeast states experienced an acute outbreak. While the overall trend in the transmission index was downward between April and September (as shown in Figure 6), the normalized distribution of the index among states was similar, as shown in Figure 8. For instance, Illinois continued to have a high index relative to other states. 

### 4.3. Sensitivity Analysis

This section evaluates the model under scenarios with different trip flows as the input for our SEIRD compartmental model. Since the trip between two states is estimated by a ratio of out-of-state flow based on the gravity model, the variation of the power-law parameters α and γ in the ratio $\frac{\frac{G_{j}^{\alpha }}{D_{ij}^{\gamma }}}{\sum_{k}\frac{G_{k}^{\alpha }}{D_{ik}^{\gamma }}}$, representing the proportion of the total trips from region i to region j, could affect the trip distribution from one state to all other states. Because real trip flow data from region i to region j was not available, statistical estimation is unavailable to fit the parameters α and γ. We have instead analyzed the scenarios of setting α at values of 0.5, 1, 1.5, and 2, respectively, while keeping γ fixed at 2, as well as assigning γ to be 0.5, 1, 1.5, and 2 with α fixed at 1 to investigate the model output and the state with the most substantial change in the transmission import index. The number of trips leaving a state is fixed in all cases and defined by our input data, as mentioned earlier.

#### 4.3.1. Varying the GDP Power Parameter

The change in the power-law parameter for the GDP affects the distribution of trips among destinations and thus affects our model output. Since a distance power parameter around two has been widely used in previous research on infectious disease modeling [10,11,13,50,51], we fixed γ to be 2 while changing the GDP power-law parameter from 0.5 to 2 with increments of 0.5, creating four scenarios based on our model. The heat map of the transmission import index for the three example periods, March to April, May to June, and August to September, is shown in Figure 9. 

When the power-law parameter on the distance term is fixed, increasing the power-law parameter for the GDP terms increases the marginal effect of the GDP as the GDP increases. Vermont consistently shows a high import index, which can be attributed to low reported cases in all periods. With few cases, local transmission is very limited, making the state most vulnerable to infected travelers coming from elsewhere. Table 1 summarizes the mean value and standard deviation of the transmission import index among all the states as the GDP power law parameter varies. We note that the standard deviation is generally larger than the mean, indicating the distribution of transmission indexes among all the states is a long-tailed distribution. The impact of human travel on disease transmission varied widely among states due to the vast geographic expanse of the United States. Some states, such as New York, California, Illinois, and typical travel destination states, would be more impacted by human travel in disease transmission, resulting in a higher transmission import index for these states. Meanwhile, some states with fewer travelers from other states might exhibit a local transmission pattern with fewer disease cases occurring due to human travel. The relatively large standard deviation reflects the different levels of engagement of each state in transportation and human travel in the United States.

The states with the largest month-to-month change in the transmission import index over time are shown in Table 2 (largest increment) and Table 3 (largest decrement). For instance, the row “April~May” in Table 1 represents the state with the biggest increase in index between the period ending in April and the period ending in May under four scenarios. As the alpha and gamma parameters change, the emphasis placed on the GDP and distance parameters also changed, which shifted the travel patterns and the choice of travel destinations predicted by the gravity model. However, the states with the largest increments and decrements in the transmission import index remained relatively stable in the investigated period. Between these periods, North Carolina experienced the greatest increase in all scenarios, whereas New York experienced the greatest in all but one scenario. Over time, the import index both increased and decreased in various states as states were differentially affected by disease import and local transmission. Those states with the largest increments would merit the attention of public health agencies, since they were the most vulnerable and would be easily affected by population mobility. The change in index reflects relative changes in multiple factors: travel from proximate states, disease prevalence in proximate states, and disease prevalence within the state. The key factor is that vulnerability to disease import changed throughout 2020 in the United States.

#### 4.3.2. Varying the Distance Power Parameter

Variation of the power-law parameter for distance may also affect the trip distribution and the model output. We fixed the power-law parameter for GDP to be 1 while changing the power parameter for distance from 0.5 to 2 with increments of 0.5. The heat map of the transmission import index for the three example periods is shown in Figure 10.

When the power-law parameter for distance changes from 0.5 to 2, the marginal effect of distance increases as distance increases. This can have the effect of trip destinations being closer to trip origins, resulting in states with a higher number of contiguous states experiencing a greater influx of travelers. Overall, our results show a relatively steady pattern in the transmission import index when the distance power-law parameter changes, representing the model’s robustness as the parameter varies. Table 4 summarizes the mean value and standard deviation of the transmission import index among all states by time period as the distance power-law parameter varies. Similar to the situation above, the standard deviation is large relative to the mean, indicating the variation in impact of human travel on disease transmission. 

## 5. Discussion

Many factors affected the spread of COVID-19 within and between states, including health policy, environmental factors (such as crowding in dwellings and public spaces), virus strains, and personal behavior. Those factors posed challenges in accurately predicting and explaining the disease dynamics with compartmental modelling. Research on the interaction between population mobilities and the COVID-19 pandemic using compartment models in other countries, such as China [50,52], Italy [51], and Singapore [53], identified limitations in capturing disease dynamics with the intervention of public health measures and personal behavior. During the summer of 2020, large gatherings at festivals and political gatherings occurred in some states, causing travel and local disease transmission. State and local policies surrounding work, schools, stores, restaurants, and masking also varied. The effectiveness of public health interventions on the disease dynamics varied depending on the timing, duration, and level of enforcement. As shown in Table 2, the states that experienced the most substantial increments in the transmission import index during the later spring of 2020 were North Carolina and New York, while during the summer and the fall, New York and Connecticut experienced the largest increase in the index. 

For Table 3, the states with the largest decrements in the transmission import index during the early spring of 2020 included New York and California. In summer and fall, the states with most substantial decrements are Illinois, California, and North Carolina. The change in index could be related to travel associated with large gatherings, which potentially contributed to the increase in disease risk, as well as disease control policies related to travel, which may explain the decrement in disease transmission risk. For instance, North Carolina relaxed its stay-home order on May 8th, thereby elevating the disease risk and potentially contributing to the increase in the transmission import index in May. In the case of New York, festivals held in mid-March could be associated with the uptick in the transmission import index, as individuals might have traveled into New York to attend those events. For both New York and Connecticut, the quarantine orders issued in late June for travelers from states with high coronavirus rates may be linked to the rise in the transmission import index during that period. 

Massachusetts’ reopening in June may also be connected to the change in the transmission import index from May to July. In the case of California, well-enforced disease control policies may be related to the decreased transmission import index in the summer and fall of 2020. The quarantine order for 15 states issued in Chicago in July could be a response to the increasing flow of travelers into Illinois in June and July, as indicated by the heightened transmission import index during that period. Though individuals were never prohibited from traveling between states, airlines and other carriers reduced operations, as fewer people elected to travel. The socioeconomic conditions of a state could also relate to the willingness of individual travel. These various factors affected both local community transmission and the spread of disease between regions, as well as their relative risk. 

The SARS-CoV-2 strain circulating during the study period also affects disease transmission and severity. For policy makers, it is crucial to understand the effects of those factors on disease spread to balance the planning of social events and pandemic control. Events that attract out-of-region visitors might risk new outbreaks, but if the disease is already prevalent, such events will be more consequential for community spread. General travel restrictions are important when regions do not yet have outbreaks, but once the disease is prevalent, travel restrictions are less consequential. By analyzing the potential impact of policies and social events on infectious disease transmission, deeper understanding could be derived to help with future decision-making and to guide public health policy making.

## 6. Conclusions

The COVID-19 pandemic has revealed the importance of understanding the risk of transmission of infectious diseases, particularly in periods when public health policies and recommendations are evolving. Investigation of disease transmission through spatial interaction, particularly region-level travel, provides insights into the complex dynamics that govern the spread of infectious diseases. By developing a multi-regional dynamic model with spatial interaction, we have captured the relationship between local transmission of disease within regions and the spread of disease from one region to another by travel. 

The transmission import index, which combines the local disease transmission and the potential for infectious travelers to spread diseases to new regions, is an important metric for assessing the risk of disease transmission between regions. By identifying high-risk areas, appropriate interventions, such as travel restrictions or quarantine measures, can be enacted to control disease spread and protect public health. During the pandemic, guidelines and restrictions were frequently modified in response to case and death data, as well as emerging scientific evidence. The ability to adapt and tailor public health responses based on the specific transmission dynamics of a disease—comparing local transmission versus disease import—is crucial for effective mitigation strategies. A wide range of factors, including public health actions and large public gatherings, may affect not just the local transmission of disease but its spread to other regions. By examining the interplay between these factors, we can better inform future decision-making processes and guide public health policy as to the effectiveness of alternative interventions.

In conclusion, this paper has shed light on the role of spatial interaction via travel in disease transmission and the importance of understanding these dynamics for effective control. The methods and findings presented here can serve as a foundation for future research, policy development, and public health interventions aimed at mitigating the impact of infectious diseases both within and between regions. While our analysis focused on the United States, the same methodology could be applied internationally. Because nations can directly control travel across their borders, they have greater capacity to reduce disease export and import, when data demonstrate that such actions are merited.

## Figures and Tables

**Figure 1 ijerph-21-00643-f001:**
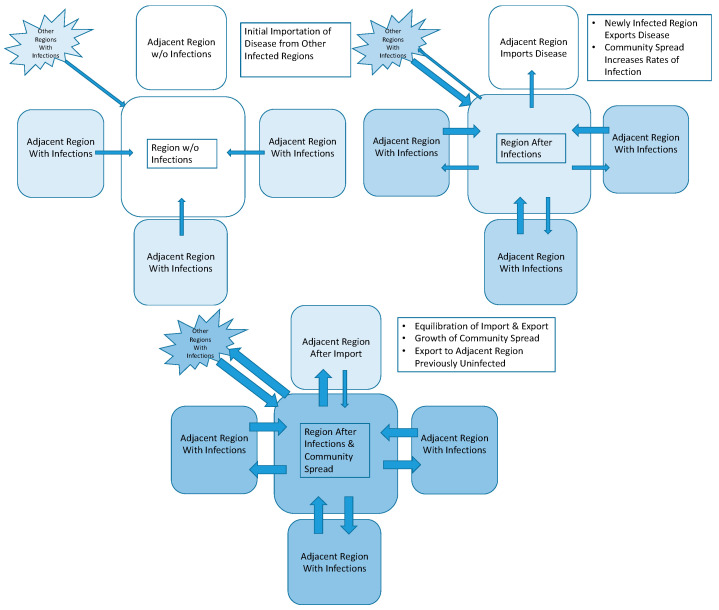
Three phases of disease spreading among regions (blue shading reflects prevalence of disease in regions, with darker shade showing greater prevalence).

**Figure 2 ijerph-21-00643-f002:**
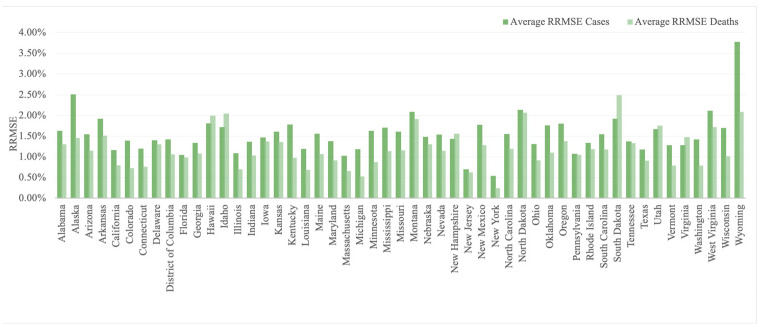
Average RRMSE for cases and deaths over 7 months among 50 states (α = 1, γ = 2).

**Figure 3 ijerph-21-00643-f003:**
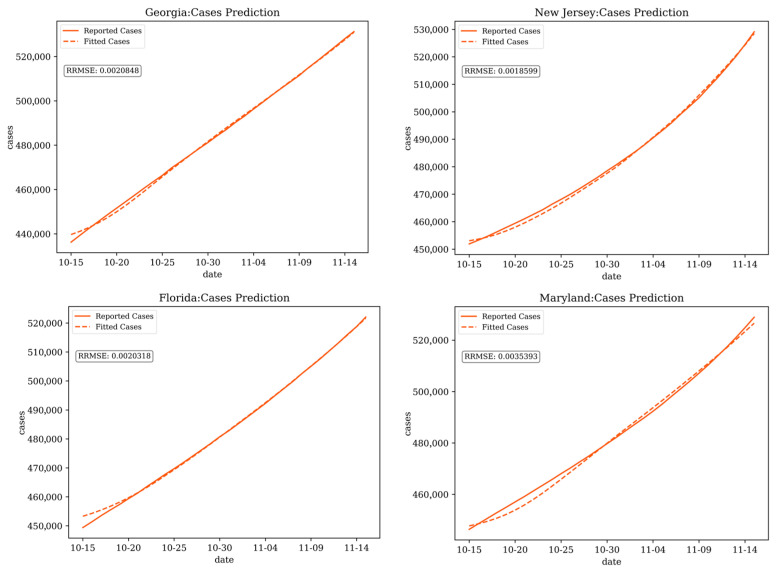
Fitting for cases in Georgia, New Jersey, Florida, and Maryland (α = 1, γ = 2).

**Figure 4 ijerph-21-00643-f004:**
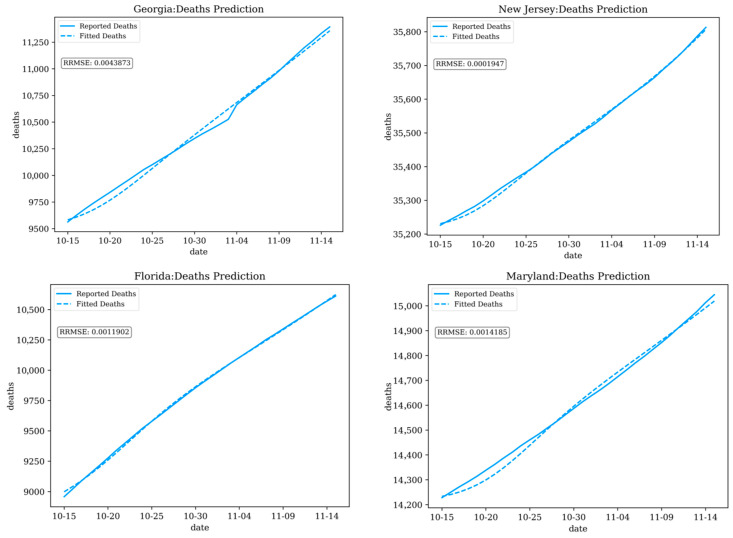
Fitting results for deaths in Georgia, New Jersey, Florida, and Maryland (α = 1, γ = 2).

**Figure 5 ijerph-21-00643-f005:**
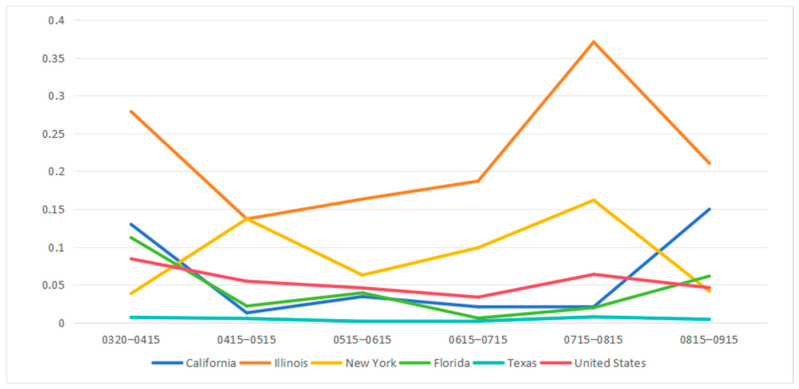
Transmission import index for six states in the US from 20 March 2020 to 15 September 2020 (α = 1, γ = 2).

**Figure 6 ijerph-21-00643-f006:**
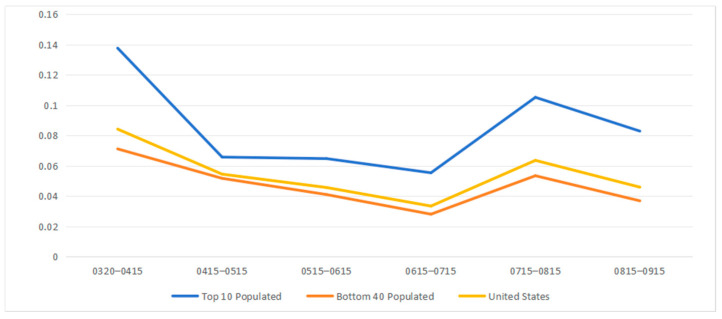
Average transmission import index for top 10 populated states and bottom 40 populated states in the US from 20 March 2020 to 15 September 2020 (α = 1, γ = 2).

**Figure 7 ijerph-21-00643-f007:**
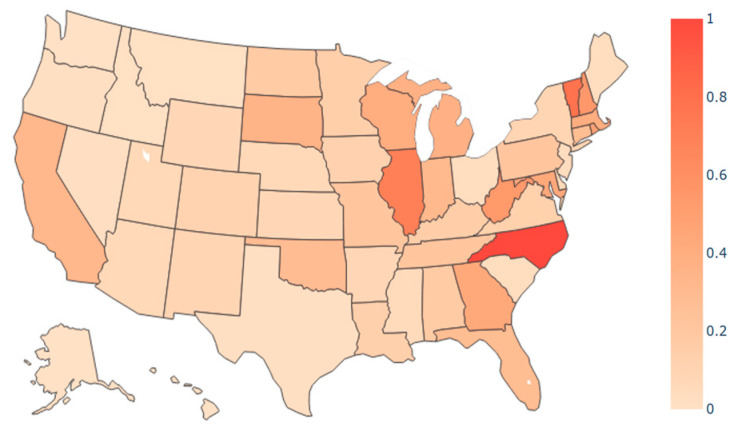
Transmission import index heat map in the US from 20 March 2020 to 15 April 2020 (α = 1, γ = 2).

**Figure 8 ijerph-21-00643-f008:**
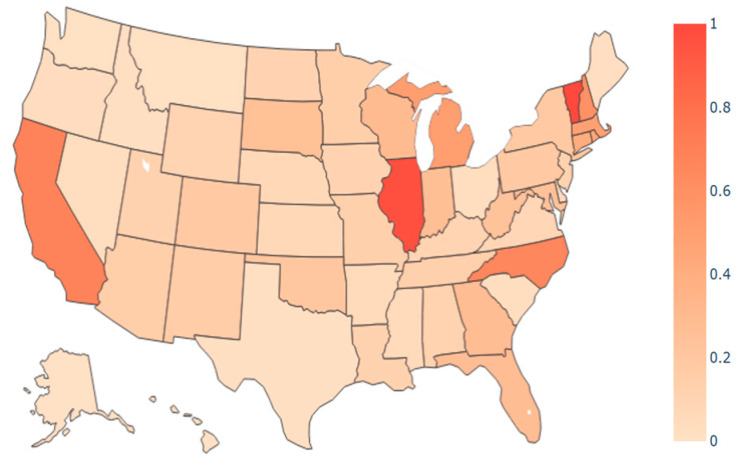
Transmission import index heat map in the US from 15 August 2020 to 15 September 2020 (α = 1, γ = 2).

**Figure 9 ijerph-21-00643-f009:**
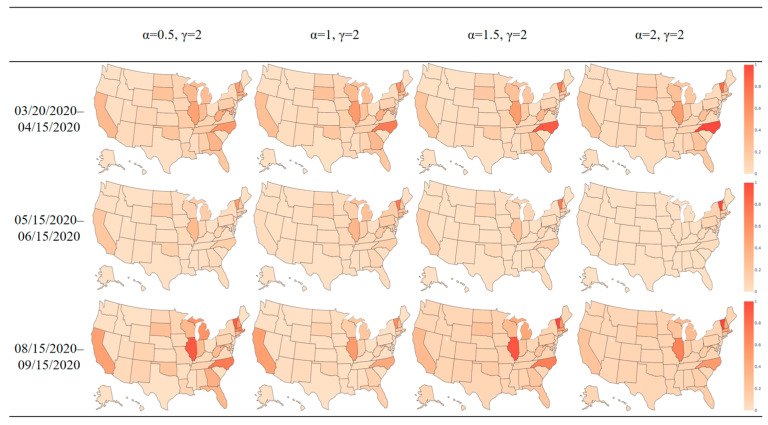
Transmission import index heat map in the US for three time periods with a changing GDP power-law parameter.

**Figure 10 ijerph-21-00643-f010:**
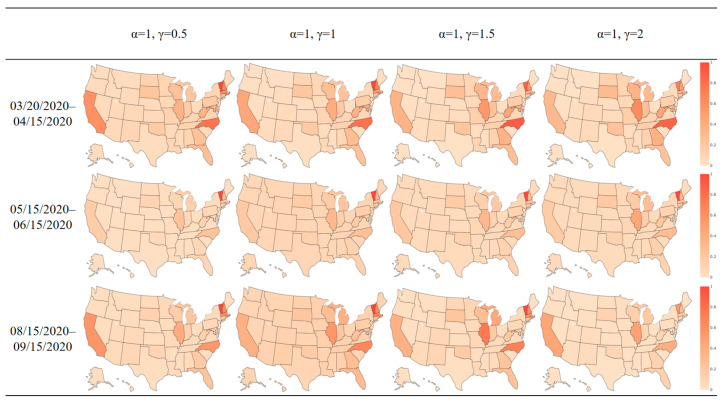
Transmission import index heat map in the US for three time periods with a changing distance power-law parameter.

**Table 1 ijerph-21-00643-t001:** Average and standard deviation of transmission import index among all states with varying alpha.

	α = 0.5, γ = 2	α = 1, γ = 2	α = 1.5, γ = 2	α = 2, γ = 2
20 March 2020– 15 April 2020	(0.0924, 0.0784)	(0.0842, 0.0868)	(0.0746, 0.0952)	(0.0678, 0.103)
15 May 2020– 15 June 2020	(0.0384, 0.0410)	(0.0457, 0.0688)	(0.0318, 0.0552)	(0.0239, 0.0813)
15 August 2020– 15 September 2020	(0.107, 0.100)	(0.0459, 0.0506)	(0.0729, 0.104)	(0.0520, 0.0861)

**Table 2 ijerph-21-00643-t002:** States with the largest increments in the transmission import index.

	α = 0.5, γ = 2	α = 1, γ = 2	α = 1.5, γ = 2	α = 2, γ = 2
**April–May**	North Carolina	North Carolina	North Carolina	North Carolina
**May–June**	Oklahoma	New York	New York	New York
**June–July**	Vermont	Vermont	Oklahoma	Florida
**July–August**	New York	New York	California	Massachusetts
**August–September**	New Jersey	Connecticut	Connecticut	Connecticut

**Table 3 ijerph-21-00643-t003:** States with the largest decrements in the transmission import index.

	α = 0.5, γ = 2	α = 1, γ = 2	α =1.5, γ = 2	α = 2, γ = 2
**April–May**	New York	New York	New York	Vermont
**May–June**	California	California	Illinois	California
**June–July**	Illinois	Illinois	Massachusetts	Illinois
**July–August**	California	North Carolina	Illinois	Illinois
**August–September**	Massachusetts	California	North Carolina	North Carolina

**Table 4 ijerph-21-00643-t004:** Average and standard deviation of transmission import index among all states with varying gamma.

	α = 1, γ = 0.5	α = 1, γ = 1	α = 1, γ = 1.5	α = 1, γ = 2
**20 March 2020–** **15 April 2020**	(0.0835, 0.103)	(0.0816, 0.0973)	(0.0869, 0.0937)	(0.0842, 0.0868)
**15 May 2020–** **15 June 2020**	(0.0359, 0.0618)	(0.0378, 0.0678)	(0.0396, 0.0655)	(0.0457, 0.0688)
**15 August 2020–** **15 September 2020**	(0.0703, 0.0943)	(0.0830, 0.108)	(0.0880, 0.104)	(0.0459, 0.0506)

## Data Availability

The datasets generated and/or analyzed during the current study are not publicly available but are available from the corresponding author on reasonable request.

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
