# Peer review of "Spatial Interaction Analysis of Infectious Disease Import and Export between Regions"

_ijerph, 2024, doi:10.3390/ijerph21050643_

Round 1

Reviewer 1 Report

Comments and Suggestions for Authors

The study is extremely interesting and brings important data from an epidemiological point of view. If the study is well described, the discussions are extremely brief, comparative data with similar studies carried out in other countries are not presented, is not be describe  the possible limits, that may exist, limits that may be related to the way in which the authorities intervened, the socio-economic conditions of the state, or the type of strain of SARS-CoV-2 that we know had different degrees of virulence and speed of spread.

the conclusions are not very well structured, and they do not bring accurate conclusions about what is important to remember from the pandemic period regarding the risk of transmission of communicable diseases in a period in which the recommendations and restrictions were periodically modified, and the way in which it the transmission impact is important to conclude, more than the general mode.

Author Response

Response to Reviewer #1 Comments

Manuscript ID: ijerph-2947217

Title: Spatial Interaction Analysis of Infectious Disease Import and Export Between Regions

Dear Reviewer,

We appreciate your insightful comments and suggestions, which have helped enhance the clarity and depth of our manuscript. Below, we address each of your comments specifically, with the revised manuscript attached.

Comment 1: The reviewer noted that comparative data with similar studies carried out in other countries are not presented.

Response:

We acknowledge the reviewer's observation regarding the absence of comparative data from studies in other countries. The unique aspect of our research lies in the quantitative analysis of external transportation impacts on local transmission dynamics, specifically through the newly introduced transmission import index. Existing studies in other regions like China, Italy, and Singapore have indeed utilized transportation data for modeling disease dynamics. However, these studies employ different methodologies that do not directly correlate with the index we have developed, making direct comparisons challenging. We have cited these studies in the discussion section to acknowledge their methodologies which offer an alternative to  our  novel approache.

Comment 2: The reviewer has expressed concerns about potential limitations related to how authorities intervened, socio-economic conditions, and the virulence and spread speed of different SARS-CoV-2 strains.

Response:

Thank you for highlighting the importance of considering various influencing factors, such as intervention strategies, socio-economic conditions, and virus strain differences. We have expanded our discussion to better address these complexities. The revised manuscript now includes a detailed analysis of how these factors might influence the disease dynamics and the effectiveness of public health policies, as evidenced by variations in the transmission import index across different states. This revision helps clarify the implications of these factors for understanding and managing disease spread.

Comment 3: The reviewer mentioned that the conclusions are not well structured and lack precise insights into the importance of various factors during the pandemic, especially given the changing recommendations and restrictions.

Response:

We appreciate your feedback on the structure and depth of our conclusions. In response, we have reorganized and refocused the conclusions section to better emphasize the dynamic nature of disease transmission amid changing public health guidelines and restrictions. The revised section now highlights critical insights on how the interplay of various factors affects the spread of communicable diseases and the implications for future public health strategies. This change aims to provide a clearer, more impactful synthesis of our findings and their relevance to pandemic response planning.

Thank you for your valuable feedback. We hope we have addressed all your comments thoroughly and appreciate your guidance throughout the revision process.

Sincerely,

Mingdong Lyu, Kuofu Liu, Randolph W. Hall

Reviewer 2 Report

Comments and Suggestions for Authors

My comments in the PDF file

Author Response

Response to Reviewer #2 Comments

Manuscript ID: ijerph-2947217

Title: Spatial Interaction Analysis of Infectious Disease Import and Export Between Regions

Dear Reviewer,

Thank you for your detailed and constructive feedback. We have addressed each of your comments to improve the manuscript, as detailed below. The revised manuscript is uploaded as attachment.

Comment 1-6: Minor typographical and punctuation errors in lines 86, 177, 253, 354, 359, and 518.

Response:

Thank you for highlighting these typographical errors. We have corrected each one as follows:

Line 86: Changed "model have been" to "model has been".

Line 177: Changed "Such as model" to "Such a model".

Line 253: Changed "Where" to lowercase and added a comma after equation 1 as suggested.

Lines 354 and 359: Corrected the errors in the references.

Line 518: Added a missing period at the end of the sentence.

Comment 7: The reviewer has requested a deeper analysis of Tables 2 and 3, expressing concern about the clarity of conclusions drawn from the data given varying parameters α and γ.

Response:

We appreciate your valuable feedback on Tables 2 and 3. In response, we have expanded our discussion in the manuscript to clarify how the variations in the parameters α and γ influence the transmission import index and its implications for disease spread across states. We have included additional details on the stability of the largest increments and decrements across states under different scenarios, highlighting the relative changes in factors such as proximity to other states and local disease prevalence, which contribute to the observed trends. This expanded discussion aims to provide a clearer understanding of how changes in parameters impact the identified states and the implications for public health interventions.

Comment 8: Suggestion to modify the presentation of statistical data in Tables 1 and 4 to prevent misinterpretation as confidence intervals.

Response:

We have modified the presentation of statistical data in Tables 1 and 4 as suggested. The values are now presented in a tuple format (mean, standard deviation) instead of using the ± notation, to avoid any misinterpretation as confidence intervals. For example, the value "0.0924 ± 0.0784" has been changed to "(0.0924, 0.0784)" to clearly indicate mean and standard deviation.

Comment 9: Concerns regarding the high variability indicated by the coefficient of variation and its implications for data reliability and conclusions.

Response:

We acknowledge the concerns regarding the high variability and its potential implications. The large standard deviation relative to the mean, as noted, is indeed indicative of significant variance in state-level transportation and local transmission rates among the states, rather than outliers. This variability is reflective of the complex and diverse impact of human mobility on disease transmission dynamics across different regions. We have added a detailed explanation to the discussion section of our manuscript, highlighting how variations in public health interventions, timing, and enforcement levels across different states contribute to this observed variability. This addition will help clarify that the high variability is an inherent feature of the data reflecting real-world complexities rather than a limitation in data quality or analytical methodology.

We hope these amendments satisfactorily address the concerns raised by the reviewer. We appreciate the opportunity to enhance our manuscript and thank you for your constructive comments.

Sincerely,

Mingdong Lyu, Kuofu Liu, Randolph W. Hall
